# Identification of Key Genes Associated with Heat Stress in Rats by Weighted Gene Co-Expression Network Analysis

**DOI:** 10.3390/ani13101618

**Published:** 2023-05-12

**Authors:** Fan Zhang, Jinhuan Dou, Xiuxin Zhao, Hanpeng Luo, Longgang Ma, Lei Wang, Yachun Wang

**Affiliations:** 1College of Animal Science and Technology, China Agricultural University, Beijing 100193, China; 2College of Animal Science and Technology, Beijing University of Agriculture, Beijing 102206, China

**Keywords:** adrenal glands, genetic marker, heat stress, RNA-seq, WGCNA

## Abstract

**Simple Summary:**

Understanding the mechanisms of heat stress is increasingly important due to global warming. However, these mechanisms are poorly understood. Here, we aimed to investigate the mechanisms underlying heat stress in rats. We identified key genes associated with rectal temperature and adrenal levels of dopamine, norepinephrine, epinephrine, and corticosterone in heat-stressed rats. In particular, methyltransferase 3, poly(ADP-ribose) polymerase 2, and zinc finger protein 36-like 1 were found to be associated with the heat stress response. These genes may be candidate genes involved in the regulation of heat stress. Our findings provide new insights into the molecular mechanisms driving heat stress in rats and might help guide future research into heat stress in mammals.

**Abstract:**

Heat stress has been a big challenge for animal survival and health due to global warming. However, the molecular processes driving heat stress response were unclear. In this study, we exposed the control group rats (*n* = 5) at 22 °C and the other three heat stress groups (five rats in each group) at 42 °C lasting 30, 60, and 120 min, separately. We performed RNA sequencing in the adrenal glands and liver and detected the levels of hormones related to heat stress in the adrenal gland, liver, and blood tissues. Weighted gene co-expression network analysis (WGCNA) was also performed. Results showed that rectal temperature and adrenal corticosterone levels were significantly negatively related to genes in the black module, which was significantly enriched in thermogenesis and RNA metabolism. The genes in the green-yellow module were strongly positively associated with rectal temperature and dopamine, norepinephrine, epinephrine, and corticosterone levels in the adrenal glands and were enriched in transcriptional regulatory activities under stress. Finally, 17 and 13 key genes in the black and green-yellow modules were identified, respectively, and shared common patterns of changes. Methyltransferase 3 (*Mettl3*), poly(ADP-ribose) polymerase 2 (*Parp2*), and zinc finger protein 36-like 1 (*Zfp36l1*) occupied pivotal positions in the protein–protein interaction network and were involved in a number of heat stress-related processes. Therefore, *Parp2*, *Mettl3,* and *Zfp36l1* could be considered candidate genes for heat stress regulation. Our findings shed new light on the molecular processes underpinning heat stress.

## 1. Introduction

Heat stress is defined as the accumulation of the body’s non-specific responses to a high-temperature environment [1]. Heat stress response relies on the regulatory activities of the hypothalamic–pituitary–adrenal (HPA) axis, hypothalamic–pituitary–thyroid (HPT) axis, and hypothalamus–pituitary–gonadal (HPG) axis, which helps the body defend against heat stress stimulation [2]. Previous studies have found that acute or chronic heat stress can alter physiological and biochemical signs at the cellular, systemic, and organismal levels in animals [3,4], involving neuroendocrine regulation [5], oxidative stress responses [6,7], and basic metabolism [8]. However, the molecular mechanisms underlying heat stress remain unclear. Heat stress has become a big concern as global temperatures rise, endangering animal survival and health. Therefore, there is an urgent need to illustrate the regulatory mechanism underlying the heat stress reaction. The development of next-generation sequencing technology [9] has made it possible to identify heat stress-responsive genes and transcripts more accurately and rapidly at the transcriptome level. This technology facilitates in-depth analysis of the molecular mechanism of heat stress [10,11].

The liver and adrenal glands are critical tissues for heat stress studies. The liver is a crucial metabolic organ that regulates numerous biological processes in response to heat stress [12]. The adrenal glands are involved in the creation and release of several hormones associated with heat stress, such as glucocorticoids, which maintain equilibrium in the body [13]. Furthermore, heat stress causes transcriptional, metabolic, and protein alterations in the adrenal glands [13] and liver [12,14] by stimulating the HPA, HPT, and HPG axes. Activation of the HPA axis is particularly essential. In response to stressors, secretagogues are released into the anterior pituitary, prompting the secretion of adrenocorticotropic hormone. This activates adrenocortical cells, which then manufacture and release glucocorticoids, such as corticosterone. The corticosterone level is a typical metric used for assessing heat stress reactions. Heat stress can also stimulate the sympathetic-adrenal medulla system and enhance catecholamine release, causing the levels of catechol hormones such as adrenaline, norepinephrine, and dopamine to rapidly rise. Masaki et al. [15] found that norepinephrine and epinephrine concentrations in human plasma increase in response to heat stress.

To investigate potential genetic markers associated with heat stress response, the gene expression data from adrenal glands and liver tissues [9] along with 14 phenotypes were used to perform weighted gene co-expression network analysis (WGCNA) [16]. The 14 phenotypes included rectal temperature (Tc), levels of corticosterone (adrenal_CORT), dopamine (adrenal_DA), epinephrine (adrenal_E), and norepinephrine (adrenal_NE) in the adrenal glands, as well as levels of corticosterone (liver_CORT), dopamine (liver_DA), epinephrine (liver_E), and norepinephrine (liver_NE) in the liver. We also examined catalase (blood_CAT), lactic acid (blood_LA), adrenocorticotropic hormones (blood_ACTH), growth hormones (blood_GH), and prolactin (blood_PRL) in the blood tissues. We screened out functional modules related to heat stress by WGCNA analysis and combined functional enrichment analysis, protein–protein interaction (PPI) networks, and short time-series expression miner (STEM) analysis, as well as conducted a phenome-wide association study (Phe-WAS) to identify key genes in heat stress response.

## 2. Materials and Methods

### 2.1. Data Sources

The data used in this study were derived from our previously published data [9]. Twenty rats were used to build thermal animal models under various environmental conditions. Five rats were housed at 22 ± 1 °C (control group, CT; *n* = 5), and three groups of five rats were placed under three different environmental conditions: 42 °C for 30 min (H30, *n* = 5), 60 min (H60, *n* = 5), or 120 min (H120, *n* = 5). In a preliminary experiment, we assessed thermosensitivity in different tissues of rats treated for varying lengths of heat stress time (30 min, 60 min, and 120 min) using receiver operating characteristic (ROC) analysis. We found that the area under the curve (AUC) was greater than 0.80 for different heat stress times, indicating that the rats were in a state of heat stress at these time points. However, it is possible that the rats experienced different degrees of heat stress at different times. Therefore, we chose these three time points for our experiment. The Tc of rats was detected using the electronic clinical thermometer with a precision of ±0.1 °C (MC-347, Omron Corporation, Kyoto, Japan). The adrenal glands, liver, and blood samples were collected from each rat. Biochemical indicators in the adrenal glands, liver, and blood tissues were examined, and gene expression levels in the adrenal glands and liver tissues were analyzed. Briefly, the concentrations of dopamine, noradrenaline, epinephrine, and corticosterone in the adrenal glands, liver, and blood tissues were determined using enzyme-linked immunosorbent assays. The Trizol method was used to extract RNA from the adrenal glands and liver, and RNA quality was evaluated via NanoDrop 2000, as described by Dou et al. [17]. On the Illumina^®^ HiSeq 2000 platform (San Diego, CA, USA), an RNA sequencing (RNA-seq) library was built and sequenced, yielding 150 bp paired-end reads. Each sample’s gene expression data were normalized using the reads per kilobase of exon model per million mapped reads (FPKM) method. Differential expression analyses were performed using a *t*-test. Genes in the liver and adrenal glands that met the criteria *p* ≤ 0.01, false discovery rate (FDR)-adjusted *p* = 0.05, and |fold change (FC)| > 2 were regarded as differentially expressed genes (DEGs) [18]. The RNA-seq datasets were released on the Sequence Read Archive at the National Center for Biotechnology Information (BioProject accession number PRJNA690189).

### 2.2. WGCNA

WGCNA (v1.12.0), implemented in the R program, was used to construct a gene co-expression network for all genes identified in the adrenal glands and liver tissues [19]. The expression levels of all genes in the 40 samples, including 20 adrenal gland samples (*n* = 5 for each treatment group: CT, H30, H60, H120) and 20 liver samples (*n* = 5 for each treatment group: CT, H30, H60, H120), were calculated using Pearson’s correlation matrices. The formula $a_{mn}={\left|c_{mn}\right|}^{\beta }$ was used to establish a weighted adjacency matrix, where $a_{mn}$ is the adjacency between genes m and n, $c_{mn}$ is Pearson’s correlation coefficient, and β is the soft-power threshold [19]. To evaluate the connectivity of each gene in the network, the weighted adjacency matrix was transformed into a topological overlap measure (TOM) matrix. A clustering dendrogram of the TOM matrix was created using the average hierarchical clustering technique. Genes with comparable patterns of expression were clustered into the same modules using the dynamic hybrid cutting method. Furthermore, using Pearson’s correlation matrices, we correlated the eigengenes of the modules (the first principle component of the corresponding expression matrix) with 13 bioindicators including adrenal_CORT, adrenal_DA, adrenal_E, adrenal_NE, liver_CORT, liver_DA, liver_E, liver_NE, blood_CAT, blood_LA, blood_ACTH, blood_GH, blood_PRL, and Tc. We choose the modules with an absolute value of correlation greater than 0.80 and a significance level less than 0.05 as functional modules. Considering that Tc is a simple and effective measure of heat stress, we subsequently selected two modules (black modules in adrenal glands and green–yellow modules in liver tissues) that were significantly related to Tc for further analysis. The visualization was completed by the WGCNA package in R software, including Figure 1A,B and Figure 2A,B.

### 2.3. Functional Enrichment Analysis of Functional Modules and Mining of Key Genes

Genes in each module were submitted to gene ontology (GO) and the Kyoto Encyclopedia of Genes and Genomes (KEGG) pathway analysis using the “clusterProfiler” in the R package to discover the biological activities and signaling pathways associated with each module [20]. The genes with the top 0.1% intramodular connectivity in significant modules were regarded as hub genes. In functional modules, genes that were identified as both hub genes and DEGs were considered key genes. Figure 1C and Figure 2C were generated using clusterProfiler and ggplot2 packages in R software. Figure 3D,E was generated by using the ggplot2 package in R-4.2.2 software, showing key genes and significant Go terms that key genes involved.

### 2.4. Short Time-Series Expression Miner (STEM) Analysis of Key Genes

STEM v1.3.13 software [21] was used to identify key genes exhibiting the same expression trend. Because there are just a few time points per dataset, the STEM clustering method selects a set of distinct and typical timed expression profiles, known as model profiles. Each gene was assigned the closest model profile by using the coefficient of correlation. Next, the computed *p*-value based on a hypergeometric distribution was applied to identify which model profiles were assigned significantly more genes. The clustering parameters were set as follows: a maximum of 50 model profiles, a maximum unit change between time points of 2, and a minimum correlation for clustering comparable profiles of >0.7 [22]. Figure 3A,B was built with the STEM v1.3.13 program.

### 2.5. Downstream Bioinformatics Analyses

To explore functions of the significant clustered genes identified via STEM analysis, we performed protein–protein interaction (PPI) network analysis (STRING, https://string-db.org/ (accessed on 15 February 2023) and a phenome-wide association study (Phe-WAS, https://atlas.ctglab.nl/ (accessed on 6 December 2022). The PPI network was investigated using four types of evidence (experimental, text mining, co-expression, and databases). The Phe-WAS was a study strategy of 3302 human phenotypes designed to discover associations between a particular SNP or gene and a wide range of traits. This strategy has proved to be effective in both recovering previously discovered genotype–phenotype relationships and finding new ones [23,24]. Figure 3C,F is drawn using STRING and Figure 4A–F is drawn using R.

## 3. Results

### 3.1. Summary of RNA-Seq Data

In total, 40 RNA-seq datasets (20 RNA-seq datasets from the adrenal and 20 RNA-seq from the liver) were obtained. After quality control and alignment analyses, 5881 and 8472 genes were identified in the adrenal glands and liver tissues, respectively [9]. In the adrenal glands (CT vs. H30, CT vs. H60, and CT vs. H120) and liver tissues (CT vs. H30, CT vs. H60, and CT vs. H120), 1501 and 1310 DEGs were identified, respectively, based on a threshold of *p* < 0.05, FDR-adjusted *p* = 0.05, and |FC| > 2.

### 3.2. Gene Co-Expression Modules Associated with Phenotypic Traits

By employing WGCNA, nine gene modules were identified in the adrenal glands samples, including eight co-expression modules and one module containing the remaining uncorrelated genes (Figure 1A,B). Six modules were identified in the liver samples, including five co-expression modules and one module containing the remaining independent genes (Figure 2A,B). In adrenal glands, the black module was significantly (*p* < 0.05) negatively correlated with rectal temperature and adrenal_CORT, with correlations of −0.9 and −0.78, respectively. In liver tissues, the green-yellow module was significantly (*p* < 0.05) positively correlated with adrenal_DA, adrenal_E, adrenal_NE, adrenal_CORT, and Tc, with correlations of 0.74, 0.78, 0.83, 0.69, and 0.77, respectively.

### 3.3. Functional Enrichment Analysis

Next, we investigated the biological function of the genes in the black and green-yellow modules. The GO terms and KEGG pathways with the top 10 black module counts are shown in Figure 1C and Figure 2C. GO enrichment analysis showed that genes in the black module were significantly involved in ncRNA metabolic processes (GO:0034660), ribonucleoprotein complex biogenesis (GO:0022613), ncRNA processing (GO:0034470), mRNA processing (GO:0006397), RNA splicing (GO:0008380), ribosome biogenesis (GO:0042254), Rrna metabolic processing (GO:0016072), Rrna processing (GO:0006364), tRNA metabolic process (GO:0006399), and mitochondrial gene expression (GO:0140053) in biological process (BP); mitochondral matrix (GO:0005759), organelle inner membrane (GO:0019866), mitochondrial protein complex (GO:0098798), nuclear speck (GO:0016607), ubiquitin ligase complex (GO:0000151), spliceosomal complex (GO:0005681), organellar ribosome (GO:0000313), mitochondrial ribosome (GO:0005761), and organellar large ribosomal subunit (GO:0000315) in cellular component (CC); and catelytic activity acting on RNA (GO:0140098), transcription coregulator activity (GO:0003712), coenzyme binding (GO:0050662), transferase activity, transferring one-carbon groups (GO:0016741), ligase activity (GO:0016874), catalytic activity, acting on a tRNA (GO:0140101), helicase activity (GO:0004386), single-stranded DNA binding (GO:0003697), aminoacyl-tRNA ligase activity (GO:0004812), and ligase activity forming carbon-oxygen bonds (GO:0016875) in molecular function (MF). The KEGG pathway analyses confirmed that genes in the black module were significantly enriched in thermogenesis (rno04714), RNA transport (rno03013), biosynthesis of cofactors (rno01240), spliceosome (rno03040), carbon metabolism (rno01200), peroxisome (rno04146), RNA degradation (rno03018), ribosome biogenesis in eukaryotes (rno03008), nucleotide excision repair (rno03420), and fatty acid metabolism (rno01212). GO enrichment analysis showed that genes in the green-yellow module were significantly involved in posttranscriptional regulation of gene expression(GO:0010608), covalent chromatin modification (GO:0016569), organic cyclic compound catabolic process (GO:1901361), RNA catabolic process (GO:0006401), mRNA catabolic process (GO:0006402), response to endoplasmic reticulum stress (GO:0034976), regulation of DNA-templated transcription in response to stress (GO:0043620), regulation of transcription from RNA plolymerase 2 promoter in response to stress (GO:0043618), bile acid and bile salt transport (GO:0015721), negative regulation of transcription from RNA polymerase 2 promoter in response to stress (GO:0097201), and ubiquitin-binding (GO:0043130). In the KEGG pathway analysis, no significant pathway was found.

### 3.4. Identification of Hub Genes in the Functional Modules and Mining of Key Genes Associated with Heat Stress

One pattern (profile 9), containing 17 key genes, was significantly enriched (*p* < 0.05) in the adrenal glands. Another pattern (profile 49), containing 13 key genes, was significantly enriched (*p* < 0.05) in the liver.

Seventeen key genes from the black module were considerably enriched in profile 9, and 13 key genes from the green-yellow module were strongly enriched in profile 49, as shown in Figure 3A,B. This indicates that key genes in the same module exhibited the same pattern of expression changes, suggesting that the module was accurately identified. In Figure 3D,E, we picked out significant Go teams in which the key gene was involved. *Mettl3* was related to RNA splicing and metabolism, methylation, and histone modification, *Parp2* was related to DNA repair, and *Zfp36l1* was involved in endoplasmic reticulum stress and transcriptional regulation under stress. In our view, these processes are all related to the presence of heat stress. Then, we performed protein–protein interaction analysis of gene sets containing key genes from the black and green-yellow modules, as shown in Figure 3C,F. In the protein–protein interaction networks, *Mettl3* and *Parp2* from the black module and *Zfp36l1* from the green-yellow module had central regulatory roles. Thus, we selected these three genes for subsequent analysis and in-depth discussion.

### 3.5. Phe-WAS of Key Genes Associated with Heat Stress in Humans

To further annotate the functions of the candidate genes in mammals, we performed Phe-WAS of human orthologs of *Parp2*, *Mettl3*, and *Zfp36l1* across 3302 human phenotypes (https://atlas.ctglab.nl/ (accessed on 6 December 2022)). The levels of expression of *Parp2* and *Mettl3* were significantly downregulated in response to heat stress, whereas *Zfp36l1* was significantly upregulated (Figure 4A,C,E). *Parp2*, *Mettl3,* and *Zfp36l1* were significantly associated with immunity, endocrine function, and metabolism (Figure 4B,D,F).

## 4. Discussion

Heat stress is a well-known phenomenon that triggers a range of bodily responses, including alterations in hormone concentrations such as dopamine, norepinephrine, epinephrine, and cortisol. Dopamine, norepinephrine, and epinephrine are all catecholamines that play significant roles in the body’s response to heat stress. Gruntenko et al. found that dopamine levels significantly rise when Drosophila suffers from heat stress [25]. Alvarez et al. also found that the norepinephrine and epinephrine levels in the blood significantly increase when cattle are exposed to heat conditions [26]. This result is consistent with the present results (attached Figure 1, Figure 2 and Figure 3). Lyte et al. suggested that catecholamine has been potentially used as an indicator of heat stress [27]. Catecholamines can increase blood flow to the skin and stimulate sweat production [28], which can facilitate heat loss and help maintain a normal core body temperature [29]. The catecholamines induce diverse physiological responses across different organs, such as reduced visceral function and digestive inhibition, enhanced cerebral blood flow, improved pulmonary gas exchange efficiency, the breakdown of glycogen to release glucose reserves, vasodilation in muscles, and elevated heart rate [30]. Although these processes can assist in resisting heat stress, they may also trigger an overactive sympathetic nervous system, which can lead to increased thermogenesis. As a result, the exact mechanism by which catecholamines operate during heat stress remains unclear.

Cortisol, an important glucocorticoid, has been shown to play the most important role in the process of heat stress in animals. When the animals suffered from heat stress, the HPA axis gets activated, which consequently increases plasma glucocorticoid concentrations in blood [31]. In the current results, cortisol showed a significant upward trend after heat stress (attached Figure 4). Cortisol has multiple functions that help animals tolerate stress. They act as vasodilators, aiding in heat dissipation and increasing blood flow [32]. Additionally, they stimulate proteolysis and lipolysis [32], which supply energy to the animal when food intake is reduced. These physiological adjustments enable the animal to cope with stress more effectively. Following exposure to heat stress, our study observed an increase in levels of dopamine, norepinephrine, epinephrine, and cortisol in the adrenal glands, while no similar increase was observed in liver tissues. With the increase in heat stress time, these hormones first increased and then decreased to the control level. This result may be related to the heat stress period. Chan et al. found that chronic heat stress does not increase dopamine, epinephrine, and norepinephrine levels in blood in rats, but is similar to that of the control group [33]. For cortisol, in response to acute heat stress, the activation of adrenocorticotrophin release in the hypothalamus is responsible for the initial increase in cortisol [34]. The later the return to normal, despite the continued heat shock, shows another reaction, most likely negative cortisol feedback [35].

To investigate the genetic and biological underpinnings of heat stress, we employed rats as model animals and implemented WCGNA. In our study, we chose two modules (black modules in the adrenal glands and green-yellow modules in the liver tissues) as functional modules. In the adrenal glands, the black module was significantly (*p* < 0.05) negatively correlated with Tc and adrenal_CORT, with correlations of 0.9 and −0.78, respectively. In the liver tissues, the green-yellow module was significantly (*p* < 0.05) positively correlated with adrenal_DA, adrenal_E, adrenal_NE, adrenal_CORT, and Tc, with correlations of 0.74, 0.78, 0.83, 0.69, and 0.77, respectively. However, in the enrichment analysis results, we could not uncover any pathways associated with the manufacture and secretion of the above hormones, indicating that the above functional modules may not be directly related to hormone secretion and activity. The process of heat stress is intricate; hormonal regulation and the function of modules may both take place at the same time. For instance, the black module affects thermogenesis, whereas the green-yellow module participates in endoplasmic reticulum stress. The top 0.1% of genes in the black and green-yellow modules were identified as hub genes, and the differentially expressed hub genes were defined as key genes. In the black module, 17 key genes were discovered, and in the green-yellow module, 13 key genes were discovered. Using STEM to cluster the key genes of the two modules, it was discovered that key genes in the same module exhibited comparable expression changes in response to changes in the duration of heat stress, indicating that the modules of key genes exhibited a consistent pattern of change. In the protein–protein interaction network, *Parp2*, *Mettl3*, and *Zfp36l1* were located in the hub region. Heat stress is known to have many effects on cells, including DNA damage [36], RNA splicing [10,37], and modification [38]. GO enrichment analysis of the black and yellow-green modules revealed that *Parp2* was involved in DNA repair, *Mettl3* was mainly involved in post-transcriptional regulation, such as RNA splicing modification and catabolism, and *Zfp36l1* was involved in transcriptional regulation during stress (Figure 3D,E).

As we see in the results of the enrichment analysis (Figure 3D), *Parp2* is involved in DNA repair. Parp2 catalyzes the synthesis of poly(ADP-ribose) (PAR) using NAD+ as a substrate [36,37], recognizes damaged DNA, and synthesizes long, branched PAR chains covalently attached either to themselves or to acceptor proteins that activate the base excision repair machinery [39,40]. Ame et al. [41] found that purified recombinant mouse Parp2 acts as a damaged DNA-binding protein in vitro and catalyzes the formation of PAR polymers in a DNA-dependent manner. Parp2 is mainly detected at a single DNA nick site and exhibits a low level of binding to undamaged DNA and double-strand breaks, according to Maria et al. [42]. In plants, stressors such as heat, light, and drought activate PARP, causing NAD+ breakdown and ATP consumption [43]. At present, many studies have demonstrated that heat stress could cause cell DNA damage [44,45]. Therefore, we initially thought that *Parp2* could be involved in heat stress regulation by mediating DNA damage repair during heat stress. However, when we looked at the expression of this gene in response to heat stress, we found that this may not be the case. Our results discovered a substantial drop in *Parp2* gene expression in rats after heat stress (Figure 4A). Heat stress may cause the inhibition of DNA damage repair by reducing *Parp2* expression, rather than cells reducing heat stress-induced damage through Parp2-mediated DNA damage repair. The current studies have shown that heat stress can inhibit key components of virtually all repair systems, including the base excision repair system [46,47,48] and nucleotide excision repair system [49,50]. So how does *Parp2* play a role in the regulation of heat stress processes? Some recent studies may shed some light on this, such as DNA methylation. Tomer et al. [51] found chicks injected intraperitoneally with Parp inhibitors showed lower body temperature, improved response to chronic heat stress, and exhibited resilience to heat stress. They found Parp inhibitors could reduce systemic DNA methylation by inhibiting the activity of methyltransferase, especially by reducing the methylation of corticotropin-releasing hormone (*CRH*) introns, resulting in decreased *CRH* expression. These results are similar to ours, and the module in which *Parp2* is located also shows significant correlations with body temperature and cortisol in the adrenal glands. The second study shows the improvement of energy utilization efficiency and the reduction in oxidative stress. Marc et al. [43] also found that Parp inhibitors could reduce NAD+ breakdown and consequently energy consumption, oxidative stress, and promote plant regeneration and repair in heat stress. This evidence shows that when exposed to heat stress, reducing the expression of the *Parp2* gene can help the body resist such stress.

Mettl3 is the catalytic component of the N6-adenosine-methyltransferase complex, which is involved in multiple processes, including RNA metabolism, gametogenesis, and DNA damage repair. *Mettl3* may also be involved in post-transcriptional regulation mechanisms, such as RNA truncation and modification, according to the results of our GO enrichment analysis (Figure 3D). In recent years, heat stress has been linked to RNA m6A methylation. Zhou et al. [38] discovered that nuclear YTHDF2 preserves the 5’UTR methylation of stress-induced transcripts, increases 5’UTR methylation in the form of m6A, and promotes cap-independent translation initiation, providing an alternative translation mechanism for selective *mRNA translation under heat shock stress. Mettl3 as the m6A “writers” is closely related to RNA m6A methylation. Yu et al. found that *Mettl3* expression was significantly downregulated and *YTHDF2* was significantly upregulated due to heat stress, and *Mettl3* knockdown resulted in significant upregulation of a variety of heat shock proteins, including HSPA1b, HSPA9, and HSPB1 [52]. Our results also prove that the expression of the *Mettl3* gene exhibited a downward trend after suffering from heat stress. Therefore, our understanding is that *Mettl3* has the potential to modulate heat stress protein expression through m6A, which could impact the heat stress response. In addition, heat stress causes changes in the activity of various antioxidant enzymes in the body, such as catalase, glutathione peroxidase, and oxide dismutase, disrupting the body’s oxidant and antioxidant balance. This results in oxidative stress, the overproduction of reactive oxygen and nitrogen species, and damage to cellular DNA. *Mettl3* is involved in the repair of UV-induced DNA damage via the RNA m6A modification pathway, according to Yang et al. [53]. In *Mettl3*-knockout cells, UV-induced cyclobutane pyrimidine adduct repair is delayed, and UV sensitivity is enhanced.

Zfp36l1 is a zinc lipoprotein that plays a role in the degradation of nuclear-transcribed mRNA. In our research, *Zfp36l1* was related to response to endoplasmic reticulum stress (GO:0034976) and regulation of DNA-templated transcription in response to stress (GO:0043620), according to the results of the GO enrichment analysis (Figure 3E). When heat stress occurs, it is accompanied by a response to endoplasmic reticulum stress through oxidative stress, resulting in the accumulation of unfolded or misfolded proteins [54]. This buildup initiates a signaling cascade known as the unfolded protein response (UPR), which is a cellular defense mechanism meant to restore ER equilibrium. The UPR response to heat stress includes the overexpression of heat shock proteins, which aid in the refolding of denatured proteins, as well as the triggering of death if the cell damage is too severe to repair (apoptosis). In reaction to oxidative stress, polo-like protein kinase 3 (Plk3) has been postulated to trigger apoptosis. For example, Deng et al. [55] discovered that plk3 played a role in oxidative stress-induced DNA damage and apoptosis, and they found that overexpressing plk3 reduced cell vitality and increased apoptosis, whereas silencing plk3 weakened the apoptotic response. Meanwhile, in another study, Zfp36l1 was able to degrade the mRNA of plk3 through the AU-rich elements in the 3’UTR region [56]. In our research, the expression of the *Zfp36l1* gene exhibited an upward trend after suffering from heat stress. Therefore, we hypothesize that overexpressed *Zfp36l1* may participate in oxidative stress and endoplasmic reticulum stress by degrading plk3, reducing autophagy under heat stress. In addition, in human research, the orthologs of *Zfp36l1* and *Zfp36l1* have been shown to influence variation in HSPA1A and HSPA1b (two important heat stress proteins) expression [57]. This may be another important pathway for the *Zfp36l1* gene to participate in heat stress.

We propose that *Parp2*, *Mettl3*, and *Zfp36l1* are candidate genes involved in the regulation of heat stress. However, this requires validation in future functional studies.

## 5. Conclusions

In this study, we performed WGCNA of RNA-seq data from adrenal glands and liver tissues to detect key genes and pathways involved in the heat stress response in rats. The black modules in the adrenal glands and green-yellow modules in the liver tissues were significantly correlated with phenotypes under heat stress, including Tc and hormone levels in the adrenal glands. Based on intramodular connectivity, differential expression analysis, PPI, and enrichment analysis, we believe that three key genes (*Parp2*, *Mettl3*, *Zfp36l1*) play an important regulatory role in the response to heat stress. Our results provide new insights into the molecular mechanisms underlying heat stress and demonstrate that integrative analyses of omics data are promising strategies for illustrating the genetic architecture underlying complex traits and diseases.

## Figures and Tables

**Figure 1 animals-13-01618-f001:**
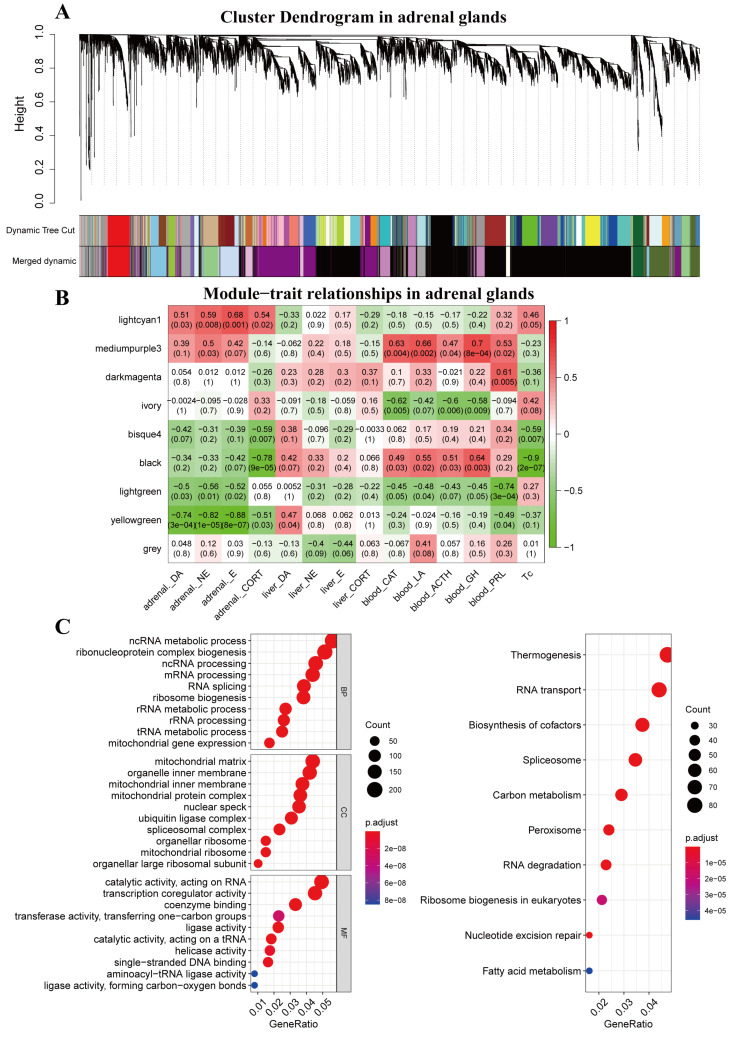
The network construction of weighted correlation network analysis and functional enrichment analysis in adrenal glands. (**A**) Clustering dendrogram of genes in adrenal glands, with dissimilarity based on the topological overlap and assigned module colors. (**B**) Modules associated with 14 bioindicators in adrenal glands. These bioindicators include DA (dopamine), NE (norepinephrine), E (epinephrine), CORT (corticosterone), CAT (catalase), LA (lactic acid), ACTH (adrenocorticotropic hormone), GH (growth hormone), PRL (prolactin), and TC (rectal temperature). The false discovery rate approach is used to adjust the statistical significance of the module–trait connections for multiple testing. (**C**) Results of the enrichment analysis of genes are in black (adrenal glands).

**Figure 2 animals-13-01618-f002:**
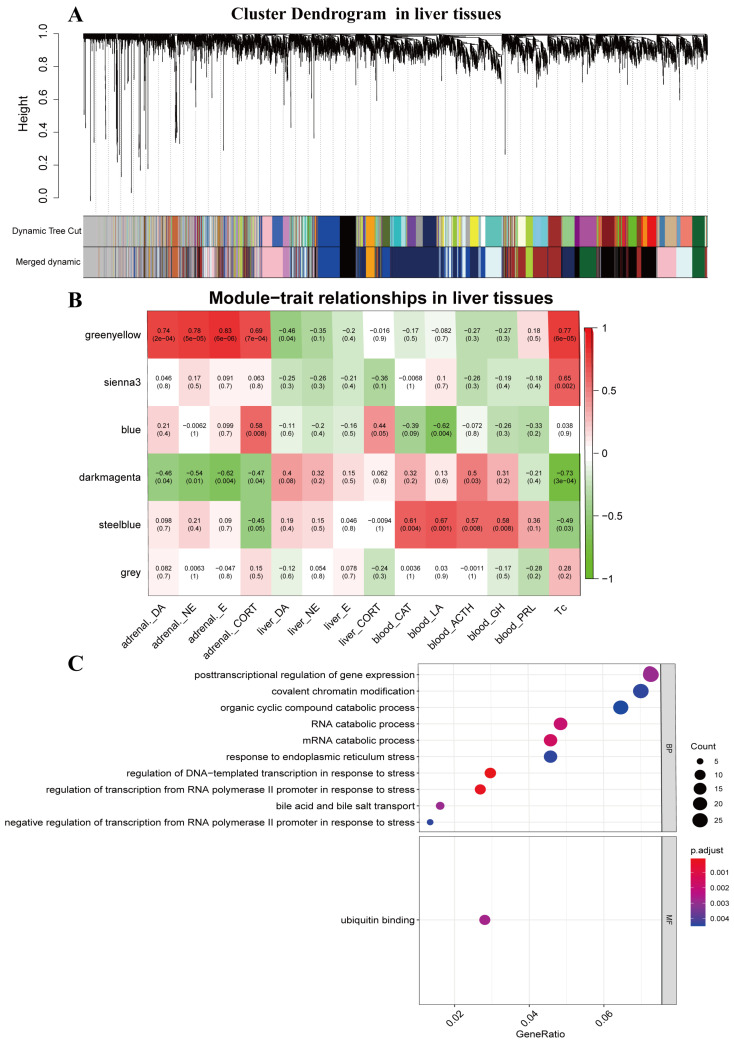
The network construction of weighted correlation network analysis and functional enrichment analysis in liver tissues. (**A**) Clustering dendrogram of genes in liver tissues, with dissimilarity based on the topological overlap and assigned module colors. (**B**) Modules associated with 14 bioindicators in liver tissues. These bioindicators include DA (dopamine), NE (norepinephrine), E (epinephrine), CORT (corticosterone), CAT (catalase), LA (lactic acid), ACTH (adrenocorticotropic hormone), GH (growth hormone), PRL (prolactin), and TC (rectal temperature). The false discovery rate approach is used to adjust the statistical significance of the module–trait connections for multiple testing. (**C**) Results of the enrichment analysis of genes are in green-yellow (liver tissues) modules.

**Figure 3 animals-13-01618-f003:**
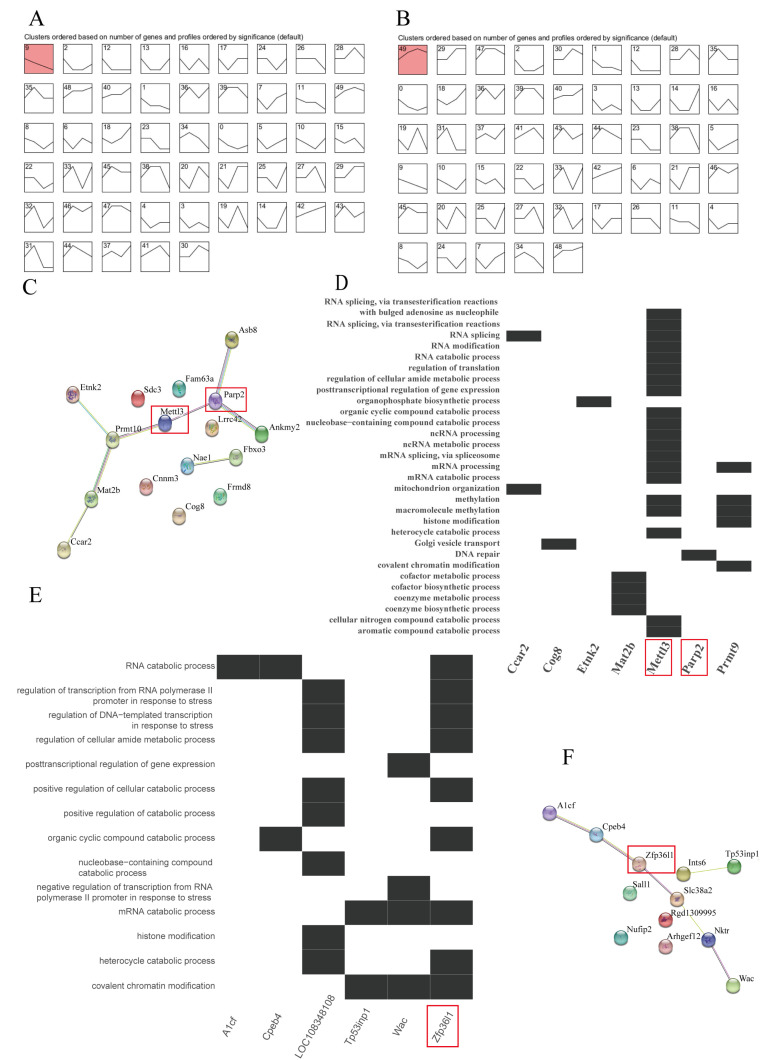
Key gene expressions and functional analysis. (**A**) Expression pattern of key genes in the black module of the adrenal glands. (**B**) Expression pattern of key genes in the green-yellow module of the liver tissues. (**C**) Protein–protein interaction network analysis (STRING database) of genes in the black module in adrenal glands. (**D**) GO terms involving key genes in the black module via enrichment analysis. (**E**) GO terms involving key genes in the green-yellow module via enrichment analysis. (**F**) Protein–protein interaction network analysis (STRING database) of genes in the green-yellow module in liver tissues.

**Figure 4 animals-13-01618-f004:**
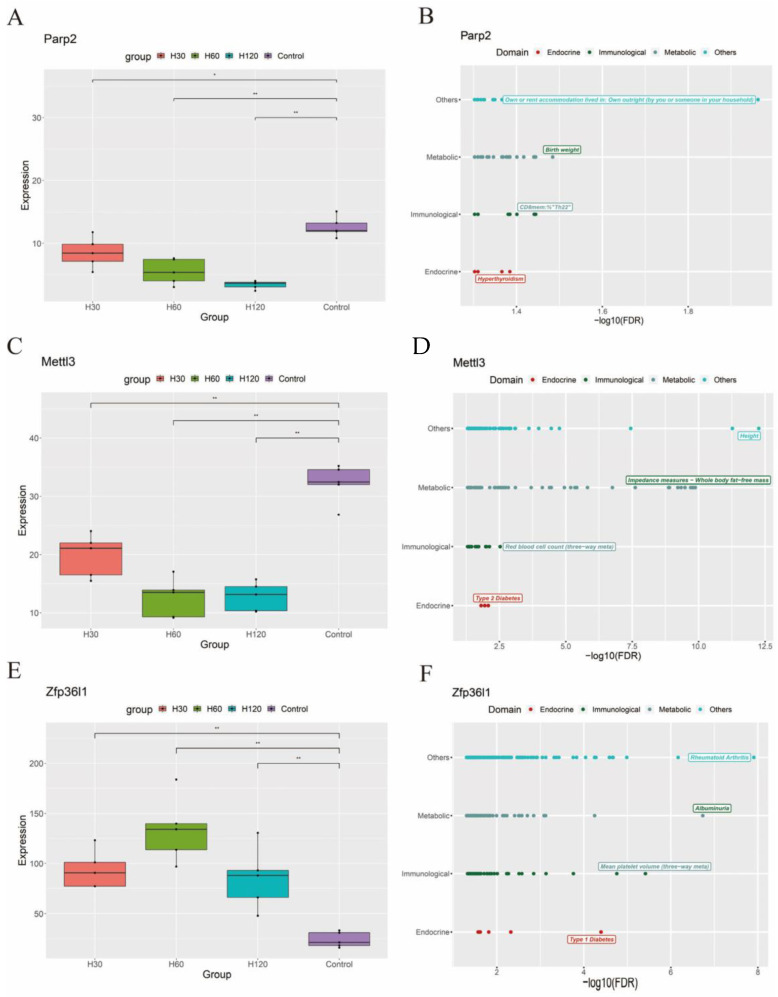
Heat stress candidate gene phenome-wide association analysis (Phe-WAS) in humans. (**A**,**C**,**E**) Boxplots depict the gene expression levels of three important genes in each of the four heat stress groups (CT, H30, H60, and H120). The significance level (*p*) was determined by a *t*-test. ** and * represent *p* < 0.01 and *p* < 0.05, respectively. (**B**,**D**,**F**) Phe-WAS results for the three key genes. The *t*-test was used to calculate *p*-values for metabolic traits and related traits.

## Data Availability

The RNA sequencing (RNA-seq) datasets were obtained from the Sequence Read Archive at the National Center for Biotechnology Information (BioProject accession number PRJNA690189).

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
