# Peer review of "Identification of Key Genes Associated with Heat Stress in Rats by Weighted Gene Co-Expression Network Analysis"

_animals, 2023, doi:10.3390/ani13101618_

Round 1

Reviewer 1 Report

The paper performed WGCNA of RNA-seq data from adrenal glands and liver tissues to detect key genes in the heat stress response in rats.

Specific comments:

(1) In 3.2, when co-expression modules of genes related to phenotypic traits were selected, why did adrenal group choose negative correlation and liver group choose positive correlation?

(2) KEGG analysis diagram of yellow-green module in 3.3 needs to be supplemented.

(3) How do we understand that the expression patterns of key genes in the same module are the same in 3.4? The pattern of significant concentration in the adrenal gland is 19. Figure 2A shows 9. Is that wrong?

(4) The text of Figure 2E and 2F is not clear, and Figure 2E and 2F were not mentioned in the article.

(5) How to reflect the correlation between Mettle and immune traits than other properties in Figure 3D?

(6)The discussion introduced too much the research background of the three genes, and did not have enough connection with their own research.

(7)How to reflect the detection of pathways associated with heat stress?

(8)Reference 17 is formatted differently from the rest.

Reviewer 2 Report

Dear authors,

The manuscript entitled “Identification of key genes associated with heat stress in rats by weighted gene co-expression network analysis” aims to investigate the mechanisms underlying heat stress in rats. The manuscript has a good simple summary, abstract and introduction, material and methods and brings interesting results, however, it needs many corrections in discussion and conclusion. The discussion superficially approaches the main focus of the study, further discussions should be made of the results. In conclusion, the other results of the study in addition to the three candidate genes must be addressed. To be accepted for publication, it needs major changes and corrections. Below are suggestions to make the manuscript suitable for publication and more informative to the reader. It seems that there is a lot of methodology and results for little discussion.

General suggestions

Abstract

-       Line 19: Correct “challange” to “challenge”;

-       Line 19: Correct “suvival” to “survival”;

-       Line 23: Add WGCNA, after Weighted gene co-expression network analysis.

-       Lines 21-23: Complete this sentence explaining that this was done in control group animals (N=5) and animals exposed to heat stress during 30 (N=5), 60 (N=5) and 120 minutes (N=5). I also miss the presentation of the number of animals used in each treatment for these analyzes in this part of the manuscript.

-       Line 24: Correct “nigative” to “negative”;

-       Was RNA sequencing done on adrenal glands, liver, and blood samples? The RNAseq wasn't done with the blood, was it? From the abstract it looks like it was made with the three tissues. Please review.

-       Keywords: arrange keywords alphabetically.

Introduction

-       Line 65: The following sentence is too long "Masaki et al. [15] found that norepinephrine concentrations in human plasma increase significantly in response to heat stress epinephrine concentrations also exhibit an insignificant upward trend.". Please rewrite it.

-       Line 69: The following sentence is too long "Using the Tc (rectal temperature), biochemical indicators including adrenal_CORT (corticosterone levels in the adrenal glands), adrenal_DA (dopamine levels in the adrenal glands), adrenal_E (epinephrine levels in the adrenal glands), adrenal_NE ( norepinephrine levels in the adrenal glands), liver_CORT (corticosterone levels in the liver), liver_DA (dopamine levels in the liver), liver_E (epinephrine levels in the liver), liver_NE (norepinephrine levels in the liver), blood_CAT (catalase levels in the blood), blood_LA (lactic acid levels in the blood), blood_ACTH (adrenocorticotrophic hormone levels in the blood), blood_GH (growth hormone hormone levels in the blood), blood_PRL (prolactin levels in the blood), as well as gene expression levels in liver and adrenal glands identified in our previous study [16], weighted gene co-expression network analysis (WGCNA) was performed, to identify candidate biomarkers based on gene set interconnectivity and relationships between gene sets and phenotypes [17]. ". Please rewrite it.

-       Lines 80-86: This part should be placed in the results/discussion/conclusion of the manuscript.

Materials and Methods

-       Line 91-93: “One group of five rats was housed at 22 ± 1 °C (control group, CT), and three groups of five rats were placed under three different environmental conditions: 42 °C for 30 min (H30, n = 5), 60 min (H60, n = 5), or 120 min (H120, n = 5).” How were the 3 times (30, 60 and 120) determined to evaluate the response to heat stress? Add this information in the manuscript.

-       Line 112-114: Explain that 20 samples were from the liver (N=5 for each treatment - CON, H30, H60 and H120) and 20 samples were from the adrenal glands (N=5 for each treatment - CON, H30, H60 and H120). It is not clear to the reader. In the abstract by the way it was written it seems that the RNAseq was done in the blood too.

-       Line 156 - Replace "genes" to “differentially expressed genes”.

Results

-       Line 152: “In total, 40 RNA-seq datasets from liver and adrenal glands were obtained.” How many from the liver and how many from the adrenal?

-       Line 167-168: The following sentence does not fit the results topic "Therefore, we concluded that these two modules play a significant part in the heat-stress response process."

-       In figure 2C and 2D other genes appear in the protein-protein interaction network analysis (STRING database), why were they not discussed?

-       In figure 2E and 2F it is not possible to clearly see the GO terms associated with key genes.

-       Why were only Parp2, Mettl3, and Zfp36l1 genes used for Phe-WAS analysis?

-       What word appears in figure 3B linked to the immunological domain of Parp2?

Discussion

-       Line 163-167: "In adrenal glands, the black module was significantly (P < 0.05) negatively correlated with rectal temperature and adrenal_CORT, with correlations of 0.9 and -0.78, respectively. In liver tissues, the green–yellow module was significantly ( P < 0.05) positively correlated with adrenal_DA, adrenal_E, adrenal_NE, adrenal_CORT, and Tc, with correlations of 0.74, 0.78, 0.83, 0.69, and 0.77, respectively. " was not discussed in the discussion of the manuscript.

-       Although you set the objective to measure the levels of hormones related to heat stress in the adrenal glands, liver, and blood, you did not discuss the results in relation to this objective.

-       - There was no discussion of the results observed between the treatments (CON, H30, H60 e H120), which in my view would be the objective of the study, discussing more deeply differences in relation to the response to heat stress between the tissues worked on as well (liver and adrenal gland). The discussion needs to be better focused on the objectives of the study.

-       Discuss figures 3A, 3C and 3E, which show the variation in gene expression between treatments. This discussion is extremely important for the manuscript.

Conclusion

Conclusions must be made considering all the results.

Reviewer 3 Report

-The abstract needs more details about the models of rat.

In methods: 

-L 93 what Tc means.

-The authors mentioned they used 20 rats distributed in 4 groups, but they did not specify how many rats and group from each rat model. 

Figures legends need more details for better presentation of data. 

Reviewer 4 Report

Fan Zhang et al. identified several genes associated with heat stress in rat adrenal gland and liver, by analyzing rectal temperature, biochemical indicators and the gene expression levels identified previously. There are still some questions which need to be elucidate.

 Comments: 1. Fig 1C, Fig 1F. More detailed descriptions for the GO or KEGG analysis are needed. The KEGG analysis for green-yellow module in liver tissues is also needed to have a comprehensive picture by combining with GO analysis. Also, the bubble charts quality needs to be improved.

2. Figure 2C, 2D, 2E, 2F. The images in these panels are also needed to be improved. The text sizes should be enlarged at least to make the information easier to get.

3. For all the charts(or plots), the authors should explain how these are generated in methods. Graphpad? Or any other software?

4. If it is possible, it is better to include some data about Parp2, Mettl3 and Zfp36l1 protein level and function under control and heat stress treatment. These will make the conclusion more believable. 

Round 2

Reviewer 1 Report

All questions have been answered. The author's answer is sincere and the current version is available for publication.

Author Response

Thank you for your comments.

Reviewer 2 Report

The suggestions regarding the abstract were considered and the abstract became more informative to the reader. The Introduction and the Material and Methods section of the manuscript are well written and informative. The results were well addressed and improved with the review. In the last review, my considerations were mainly made to improve the discussion of the results. The authors have now made a more in-depth discussion of the results taking into account the suggestions and the conclusion is much more in-depth based on the results. My corrections are now small and follow below (minor review):

- Line 69: Alter “geneti” to  “genetic”.

- Line 80: Alter “phe-nome-wide association study “ to “phenome-wide association study”.

- In the legend of the figures 1 and 2, please add the information referring to hormones (DA, NE, E, CORT, CAT...).

- Line 245: Alter “We” to “we”.

- Line 279-314: Paragraph too long. I suggest separating the text of these lines into more paragraphs (example: a paragraph from line 279-296, another paragraph to talk about cortisol - line 296-314).

- Line 297: Alter “when” to “When”.

- Line 373: Alter “Parp2” to “Parp2
